# Development and Validation of a Quick Sepsis-Related Organ Failure Assessment-Based Machine-Learning Model for Mortality Prediction in Patients with Suspected Infection in the Emergency Department

**DOI:** 10.3390/jcm9030875

**Published:** 2020-03-23

**Authors:** Young Suk Kwon, Moon Seong Baek

**Affiliations:** 1Department of Anaesthesiology and Pain Medicine, College of Medicine, Hallym University, Chuncheon Sacred Heart Hospital, Chuncheon 24253, Korea; gettys@hallym.or.kr; 2Division of Pulmonary, Allergy and Critical Care Medicine, Hallym University Dongtan Sacred Heart Hospital, Hwaseong-si 18450, Korea; 3Lung Research Institute of Hallym University College of Medicine, Chuncheon-si 24253, Korea

**Keywords:** qSOFA, infection, sepsis, machine learning, emergency department

## Abstract

The quick sepsis-related organ failure assessment (qSOFA) score has been introduced to predict the likelihood of organ dysfunction in patients with suspected infection. We hypothesized that machine-learning models using qSOFA variables for predicting three-day mortality would provide better accuracy than the qSOFA score in the emergency department (ED). Between January 2016 and December 2018, the medical records of patients aged over 18 years with suspected infection were retrospectively obtained from four EDs in Korea. Data from three hospitals (*n* = 19,353) were used as training-validation datasets and data from one (*n* = 4234) as the test dataset. Machine-learning algorithms including extreme gradient boosting, light gradient boosting machine, and random forest were used. We assessed the prediction ability of machine-learning models using the area under the receiver operating characteristic (AUROC) curve, and DeLong’s test was used to compare AUROCs between the qSOFA scores and qSOFA-based machine-learning models. A total of 447,926 patients visited EDs during the study period. We analyzed 23,587 patients with suspected infection who were admitted to the EDs. The median age of the patients was 63 years (interquartile range: 43–78 years) and in-hospital mortality was 4.0% (*n* = 941). For predicting three-day mortality among patients with suspected infection in the ED, the AUROC of the qSOFA-based machine-learning model (0.86 [95% CI 0.85–0.87]) for three -day mortality was higher than that of the qSOFA scores (0.78 [95% CI 0.77–0.79], *p* < 0.001). For predicting three-day mortality in patients with suspected infection in the ED, the qSOFA-based machine-learning model was found to be superior to the conventional qSOFA scores.

## 1. Introduction

The early recognition and prompt treatment of sepsis in the emergency department (ED) are important to improve patient outcomes [1]. Sepsis has been characterized as a systemic inflammatory response syndrome (SIRS) to infection [2]. The quick sepsis-related organ failure assessment (qSOFA) criteria [3] were introduced by the Third International Consensus Definitions for Sepsis and Septic Shock (Sepsis-3). The qSOFA scores can be used outside of the intensive care unit (ICU) for predicting mortality or ICU stay [4,5,6,7,8,9]. However, several meta-analyses of qSOFA scores have shown that these scores had a poor sensitivity for predicting in-hospital mortality [10,11,12,13,14]. Early warning scores [15,16], a tool for identifying hospitalized patients at risk of deterioration, have been proposed for predicting hospital mortality in those with suspected sepsis in the ED [7,17,18,19,20]. However, Hamilton et al. reported that the early warning scores were not accurate in predicting sepsis mortality in the ED (67% of sensitivity and 60% of specificity) [21].

Recently, various machine-learning methods for predicting outcomes more accurately have been implemented in the medical field [22,23,24]. Machine-learning models for the early identification of patients at risk for sepsis have been developed in the ICU [25] and ED settings [26,27,28]. Although these diverse machine-learning models can improve predictive accuracy for sepsis outcomes, they require excessive variables and laboratory results that may not be available in the ED. These factors can lead to poor generalizability of the machine-learning-based prognostic models.

The objective of this study was to develop and validate the prognostic performance of a qSOFA-based machine-learning model for three-day mortality prediction in patients with suspected infection in the ED.

## 2. Materials and Methods

### 2.1. Study Design

This retrospective multicenter study was conducted between January 2016 and December 2018 in four hospitals (Dongtan Sacred Heart Hospital, Kangnam Sacred Heart Hospital, Chuncheon Sacred Heart Hospital, and Hallym University Sacred Heart Hospital) of the Hallym University Medical Center (capacity of more than 3000 beds) in the Republic of Korea. Approximately 150,000 patients visited the EDs per year during the study period with a mean of 37,000 (range: 19,000–52,000) annual visits. A total of 447,926 patients visited the ED during the study period. The medical records of 23,587 patients older than 18 years suspected of having an infection were accessed and their data were analyzed.

The Institutional Review Board of the Hallym Medical Center approved the study (approval no. 2019-10-017-001) and waived the requirement for informed consent due to the retrospective nature of the analyses.

### 2.2. Data Collection and Definition

Clinical data were extracted using the clinical big data analytic solution Smart Clinical Data Warehouse (CDW) from the Hallym University Medical Center, which is based on the QlikView Elite Solution (Qlik, King of Prussia, PA, USA). It analyzes the electronic medical record (EMR) text and integrated fixed data. Using the Smart CDW, we collected the following clinical data of patients with suspected infection: demographic variables (age and sex), diagnoses at the ED, initial vital signs (systolic blood pressure, respiration rate, mental status, body temperature, and heart rate), arterial partial pressure of carbon dioxide, white blood cell count, duration of hospitalization, ICU admission, mechanical ventilation, and mortality. Mental status was evaluated by emergency specialist nurses using the alert, voice, pain, unresponsive (AVPU) scale (Appendix A). Body temperature was assessed using a tympanic ear thermometer.

Among the patients admitted to the ED, the data of those whose main diagnoses were infection-related were extracted. Infection-related diagnoses were referenced in a previous study by Rodriguez et al. [29]. Two of the authors (pulmonologist and anesthesiologist) reviewed and examined the patient data for eligibility, and ambiguous inclusions were excluded by consensus review (Appendix A). To avoid recording errors in the EMRs, we excluded systolic blood pressure, heart rate, respiratory rate, and body temperature values that were outside the ranges of 30–300 mmHg, 10–300 beats/min, 3–60 breaths/min, and 30–45°C, respectively [30]. The primary outcome was mortality within 3 days of admission to the ED and secondary outcomes were in-hospital mortality, ICU admission within 3 days of admission to ED, and ICU admission.

The qSOFA criteria consist of systolic blood pressure ≤ 100 mmHg, respiratory rate ≥ 22 breaths/min, and altered mental status [3]. We considered altered mental status as non-alert per the AVPU scale [31]. Patients were assigned one point for each criterion, and a qSOFA score of ≥ 2 was considered indicative of poor outcomes. The SIRS criteria are considered to be met if at least two of the following four clinical findings are present [2]: temperature > 38 °C or < 36 °C, heart rate > 90/min, respiratory rate > 20/min or arterial partial pressure of carbon dioxide < 32 mm Hg, and white blood cell count > 12 × 10^9^/L or < 4 × 10^9^/L or immature band cells > 10%. Additionally, the modified early warning score (MEWS) is widely used for detecting clinical deterioration in patients in the ED [16]; it is also used in prediction assessment for in-hospital mortality or ICU admission (Appendix A) [32].

### 2.3. Statistical Analysis and Machine Learning

Data on patient demographics, severity of illness scores (qSOFA, SIRS, and MEWS), diagnoses on admission, and outcome variables were analyzed using descriptive statistics. Continuous and categorical variables are expressed as the median (interquartile range) and number (percentage), respectively. We assessed the discriminatory power of each severity of illness score using the area under the receiver operating characteristic (AUROC) curves. All statistical analyses were performed using the Statistical Package for the Social Sciences (SPSS) Version 24.0 (IBM Corporation, Armonk, NY, USA).

The qSOFA-based machine-learning model was developed using the qSOFA criteria, including systolic blood pressure, respiratory rate, and mental status. A dataset was created with the variables of the 23,587 patients. We divided the dataset into training and validation test sets to prevent the model from overfitting the test set. The test set consisted of data from other hospitals that were not associated with training-validation to test the model. The data from three hospitals (Dongtan Sacred Heart Hospital, Kangnam Sacred Heart Hospital, and Hallym University Sacred Heart Hospital) were used for training-validation (*n* = 19,353) and the data from one hospital (Chuncheon Sacred Heart Hospital) as the test dataset (*n* = 4234). The training-validation set was divided into the training and validation sets at a 9:1 ratio. The overall dataset was divided as follows: 74% for the training set, 8% for the validation set, and 18% for the test set. Datasets were standardized by min-max scaling. Regarding the model algorithm, extreme gradient boosting (XGB), light gradient boosting machine (LGBM), and random forest were used. In the training dataset, deaths and ICU inpatients were much fewer than survivors and general ward inpatients, respectively. This data imbalance can bias the machine-learning models and render them inaccurate. To solve this problem, we trained the model after balancing the training dataset through the synthetic minority oversampling technique to a 1:1 ratio in the XGB and LGBM models and modeled the balanced random forest. Three models were trained with the basic hyperparameters and training sets and evaluated with a validation set to select the one with the best performance. We performed 5-fold cross validation on the training dataset and tuned the hyperparameters using grid search. The final model was validated through the test set. We evaluated models with the ROC curve and AUROC using Anaconda (Python version 3.7, https://www.anaconda.com; Anaconda Inc., Austin, TX, USA), the XGBoost package version 0.90 (https: //xgboost.readthedocs.io), the LGBM package version 2.2.3 (https://lightgbm.readthedocs.io/en/latest/Python-Intro.html), and the imbalanced-learn package version 0.5.0 (https://imbalanced-learn.readthedocs.io).

DeLong’s test was used to compare AUROCs between qSOFA scores and qSOFA-based machine-learning models [33].

## 3. Results

Characteristics of patients with suspected infection in the ED are summarized in Table 1. A total of 23,587 patients suspected of having an infection were enrolled, and their data were analyzed (Figure 1). The median age was 63 years (IQR: 43–78 years) and 46.1% were men (*n* = 10,862). Approximately 21.9% (*n* = 5173) of the patients were admitted to the ICU and 941 died in the hospital (4.0%). The most common diagnoses were respiratory (28.6%, *n* = 6736), intra-abdominal (24.1%, *n* = 5693), and urinary tract (15.4%, *n* = 3638) infection. There were significant differences in median age, severity of illness scores, ICU admission, and in-hospital mortality between the test and training-validation datasets.

Predictive performance of algorithm-specific machine learning models through validation sets are shown in Appendix A shows the cross-validation results of selected models. The AUROCs showing the performance of the qSOFA-based machine-learning model for outcome prediction are presented in Figure 2. For predicting three-day mortality among patients with suspected infection in the ED, the AUROC of the qSOFA-based machine-learning model was 0.85 (95% CI 0.83–0.86). The prognostic performance of the qSOFA-based machine-learning model for the prediction of several outcomes was as follows: 0.75 (95% CI 0.74–0.76) for in-hospital mortality, 0.79 (95% CI 0.79–0.79) for three -day ICU admission, and 0.79 (95% CI 0.78–0.79) for ICU admission. The AUROCs of the qSOFA-based machine-learning model for three-day mortality in the higher (≥2) and lower (<2) qSOFA groups were 0.76 (95% CI 0.67–0.81) and 0.77 (95% CI 0.71–0.80), respectively (Figure 3).

For predicting three-day mortality, qSOFA scores and MEWS showed better discrimination ability than the SIRS (qSOFA = 0.78 [95% CI 0.68–0.88]; MEWS = 0.77 [95% CI 0.67–0.86]; SIRS = 0.68 [95% CI 0.57–0.79]) (Table 2). The AUROC of qSOFA scores was higher for three-day mortality than for in-hospital mortality, three-day ICU admission, and ICU admission (0.71 [95% CI 0.66–0.75] for in-hospital mortality; 0.73 [95% CI 0.72–0.75] for three-day ICU admission; 0.73 [95% CI 0.72–0.75] for ICU admission).

In the prediction of outcome, the discriminatory abilities of the qSOFA scores and machine-learning models in the test set are shown in Table 3. Prediction performance of the qSOFA-based machine-learning model (0.86 [95% CI 0.85–0.87]) for three-day mortality was significantly higher than the conventional qSOFA scores (0.78 [95% CI 0.77–0.79], *p* < 0.001). Compared with the qSOFA scores, machine-learning models demonstrated a significantly higher AUROC.

## 4. Discussion

In this multicenter study, we applied three machine-learning algorithms (extreme gradient boosting, light gradient boosting machine, and random forest) using three qSOFA variables to predict three-day mortality in patients with suspected infection in the ED. The outcome predictive abilities of the qSOFA-based machine-learning model performed in the independent test set was satisfactory. Particularly, prediction performance of the machine-learning model for three-day mortality was superior to the conventional qSOFA score. Furthermore, we developed a qSOFA-based machine-learning model for predicting three-day mortality in the lower (<2) qSOFA groups that showed an acceptable AUROC.

Recently, machine-learning models have been applied for predicting diverse outcomes in the ED, e.g., cardiac arrest prediction [24], ED triage [34,35,36], prediction of hospital admission [37], identification of patients with suspected infection [27], screening of sepsis [28] or septic shock [26], and mortality prediction in patients with sepsis [38] or suspected infection [39]. Our study suggests that the ability of machine-learning models for predicting deterioration within three days of patients with suspected infection are superior to the conventional severity illness scores. Machine-learning models, which can predict sepsis outcome with high accuracy, are important because the sepsis is a medical emergency [40] and there is limited time for patient care in the ED. Horng et al. demonstrated that accurate triggering of clinical decision support will become increasingly more important as clinical decision support becomes more integrated into EMRs [27]. Cho et al. reported that a deep learning-based early warning system, which can be applied with EMRs, accurately predicted the deterioration of patients [41]. We suggest that our qSOFA-based machine-learning model incorporated with real-time clinical variables on the EMR can be utilized by physicians for making clinical decisions for treating sepsis.

The qSOFA criteria consist of systolic blood pressure (≤100 mmHg), respiratory rate (≥22 breaths/min), and altered mental status (three point each) [3]. However, the cutoffs for qSOFA can be potentially arbitrary [42]. On the other hand, our machine learning models using continuous variables may enable finer classification of dataset [43]. Machine leaning approaches are adept at handling high-order interactions between the predictors and non-linear relationships with the outcome [35]. Additionally, ensembles of decision tree methods like gradient boosting can automatically provide estimates of feature importance from a trained predictive model. Feature importance provides a score that indicates how valuable each feature is in the construction of the boosted decision trees within the model. We suggest that these factors contribute to better predictive ability of machine learning models than conventional qSOFA scores.

The qSOFA scores are recognized as a parsimonious tool for other complex and cost consuming way to screen for sepsis outcome [3]. Our findings support that the qSOFA score can be acceptable for predicting acute deterioration in patients with suspected infection (AUROC: three-day mortality = 0.78 and three-day ICU admission = 0.73). However, the results of a recent meta-analysis showed that the qSOFA scores are not useful for predicting in-hospital mortality (AUROC = 0.68) or ICU admission (AUROC = 0.65) [44]. Moreover, because of the poor sensitivity of the qSOFA scores, there is a possibility of delays in sepsis identification [45]. However, novel machine-learning models can accurately predict sepsis onset beforehand [46]. In accordance with the results of previous studies [38,39], our machine-learning models showed higher accuracy rate of mortality prediction in patients with suspected infection than the qSOFA scores. Therefore, machine-learning based prediction models may be beneficial for physicians in management of sepsis in the future.

The major strength of this multicenter study was the use of a large dataset and the successful prediction of mortality in the test set using the machine-learning models. These results are reliable because the model was tested using an independent test set that had patient characteristics different from those in training-validation.

Nevertheless, the study had several limitations due to its retrospective design. Although both authors carefully reviewed the diagnoses, the study population was difficult to be determined because the definition of suspected infection varied among studies. Additionally, we cannot confirm that the attending physician had documented the diagnosis. The indication of ICU admission may vary among different EDs. Therefore, further prospective studies are needed to examine the efficacy of these machine-learning models for predicting mortality in patients with infection in the ED setting.

## 5. Conclusions

We developed machine-learning models using three qSOFA criteria to predict three-day mortality in patients with suspected infection in the ED. The qSOFA-based machine-learning models are superior to the conventional qSOFA scores. We suggest that our qSOFA-based machine-learning models can assist physicians’ clinical decision-making for treating sepsis in the ED.

## Figures and Tables

**Figure 1 jcm-09-00875-f001:**
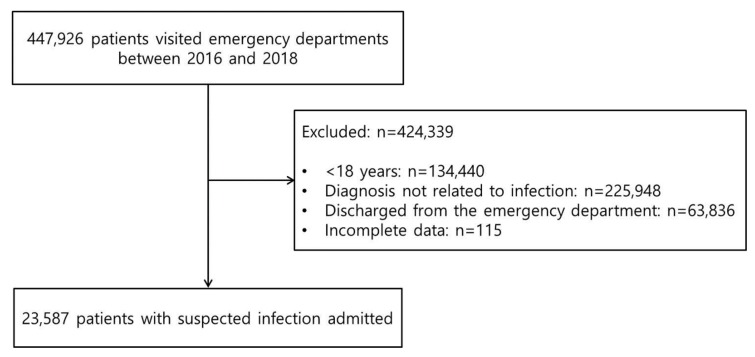
Flow chart.

**Figure 2 jcm-09-00875-f002:**
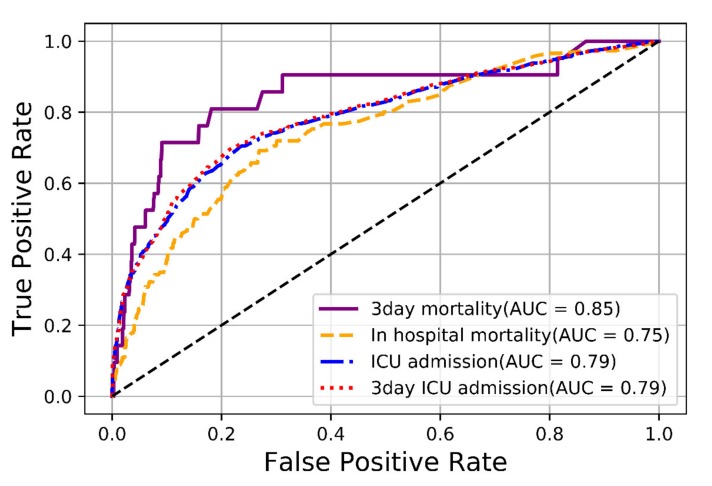
The area under the receiver operating characteristic (AUROC) curve of the machine-learning models for predicting outcomes in the test set. ICU = intensive care unit.

**Figure 3 jcm-09-00875-f003:**
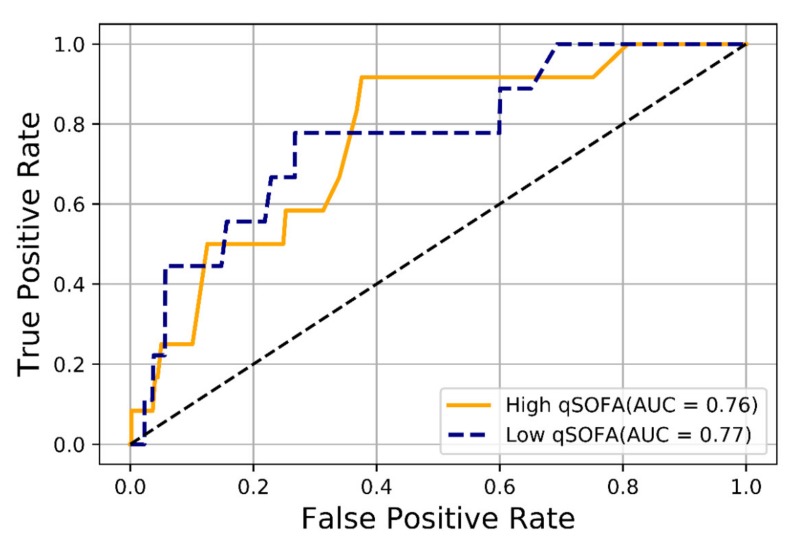
The area under the receiver operating characteristic (AUROC) curve of the machine-learning models for predicting 3-day mortality in the higher (≥2) and lower (<2) qSOFA groups. qSOFA = quick Sepsis-related Organ Failure Assessment.

**Table 1 jcm-09-00875-t001:** Characteristics of patients with suspected infection in the emergency department.

Variable	Total (*n* = 23,587)	Training-Validation Datasets (*n* = 19,353)	Test Datasets (*n* = 4234)	*p* Value
Median age (IQR)	63 (43–78)	62 (42–77)	67 (50–79)	<0.001
Male sex (%)	10862 (46.1)	8850 (45.7)	2012 (47.5)	0.035
Severity of illness scores (%)				
qSOFA ≥ 2,	4698 (19.9)	1692 (8.7)	507 (12.0)	<0.001
SIRS ≥ 2	12224 (51.8)	9960 (51.5)	2264 (53.5)	0.018
MEWS ≥ 5	5857 (24.8)	4517 (23.3)	1340 (31.6)	<0.001
Suspected infection source (%)				<0.001
Respiratory	6736 (28.6)	5437 (28.1)	1299 (30.7)	
Intra-abdominal	5693 (24.1)	4622 (23.9)	1071 (25.3)	
Urinary	3638 (15.4)	2998 (15.5)	640 (15.1)	
Hepatobiliary	1871 (7.9)	1546 (8.0)	325 (7.7)	
Otorhinolaryngological	1789 (7.6)	1481 (7.7)	308 (7.3)	
Skin or musculoskeletal	1132 (4.8)	944 (4.9)	188 (4.4)	
Gynecological	430 (1.8)	393 (2.0)	37 (0.9)	
Central nervous system	410 (1.8)	342 (1.8)	68 (1.6)	
Other or unknown	1888 (8.0)	1590 (8.2)	298 (7.0)	
Outcomes				
In-hospital mortality (%)	941 (4.0)	795 (4.1)	146 (3.4)	0.048
ICU admission (%)	5173 (21.9)	4191 (21.7)	982 (23.2)	0.029
Hospital length of stay, median (IQR), d	7 (5–12)	7 (5–12)	8 (5–13)	0.004
Mechanical ventilator use (%)	1662 (7.0)	1320 (6.9)	330 (7.8)	0.036

Values are expressed as median (interquartile range) or number (%). IQR = interquartile range; ED = emergency department; qSOFA = quick Sepsis-related Organ Failure Assessment; SIRS = systemic inflammatory response syndrome; MEWS = modified early warning score; ICU = intensive care unit.

**Table 2 jcm-09-00875-t002:** Area under the receiver operating characteristic curve for outcomes according to the severity of illness scores from the independent test set.

Variable	AUROC (95% CI)			
	3-Day Mortality	In-Hospital Mortality	3-Day ICU Admission	ICU Admission
qSOFA	0.78 (0.68–0.88)	0.71 (0.66–0.75)	0.73 (0.72–0.75)	0.73 (0.72–0.75)
SIRS	0.68 (0.57–0.79)	0.66 (0.62–0.70)	0.63 (0.62–0.65)	0.63 (0.61–0.65)
MEWS	0.77 (0.67–0.86)	0.65 (0.61–0.70)	0.69 (0.67–0.71)	0.69 (0.67–0.70)

AUROC = Area under the receiver operating characteristics; CI = confidence interval; qSOFA = quick Sepsis-related Organ Failure Assessment; SIRS = systemic inflammatory response syndrome; MEWS = modified early warning score; ICU = intensive care unit.

**Table 3 jcm-09-00875-t003:** Prediction performance of the qSOFA scores and machine-learning models in the test set.

Models	qSOFA Scores	qSOFA-Based Machine-Learning Models	
Outcomes	AUROC (95% CI)		*p* Value
3-day mortality	0.78 (0.77–0.79)	0.86 (0.85–0.87)	<0.001
In-hospital mortality	0.71 (0.69–0.72)	0.75 (0.74–0.76)	0.002
3-day ICU admission	0.73 (0.72–0.75)	0.79 (0.78–0.80)	<0.001
ICU admission	0.73 (0.72–0.75)	0.79 (0.77–0.80)	<0.001

AUROC = area under the receiver operating characteristics; CI = confidence interval; qSOFA = quick Sepsis-related Organ Failure Assessment; ICU = intensive care unit.

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
