# Peer review of "Development and Validation of a Quick Sepsis-Related Organ Failure Assessment-Based Machine-Learning Model for Mortality Prediction in Patients with Suspected Infection in the Emergency Department"

_jcm, 2020, doi:10.3390/jcm9030875_

Round 1

Reviewer 1 Report

Thank you for the opportunity to review the manuscript titled: “Development and validation of a quick Sepsis- related Organ Failure Assessment-based machine- learning model for mortality prediction in patients with suspected infection in the emergency department” This study illustrated a retrospective study on developing a machine learning model incorporating qSOFA variables, which is easy to apply for predicting short-term mortality in the emergency department.

The manuscript was well written. It simple to read and easy to understand. Sepsis is a very complex disease which need plenty of parameters in order to predict it outcome. Therefore, by using machine-learnings made it more accurate in predict sepsis outcome. Several studies have been conducted to predict disease using machine- learning showed promising result. Currently, the use of machine- learning to predict sepsis outcome is not uncommon.

Authors had justified the need to use machine learning in sepsis outcome predication. The objectives are clear and sound.

These are the comments for the authors:

Method section

How authors justify which hospital’s data set to be used as test and validation set?

Since it is commonly knowing that for big data set analyses, a ratio of 75:25 training and validation set are good enough. Could the authors explained the rational of using 9:1 ratio for training and validation set?

Author stated that the AUROC value is higher for machine-learning compare comparing the conventional scoring system by number. The authors shall provide p-value for each comparison of AUROC value against each scoring system to further validate the result.  

There is a long list of parameters were extracted from the data set. Please provide the exact data parameters that were included into the qSOFA-based machine-learning module.

Please provide a figure for the data inclusion and study flow.

Discussion section

Why is there a different of AUROC between conventional qSOFA scoring and qSOFA based-learning since the same three parameters were using in both method of analysis? What were the factors contributing to this higher AUROC for qSOFA-based machine-learning module?

qSOFA were recognized as a parsimonious tool for other complex and cost consuming way to screen for sepsis outcome (Seymour et al 2016). It is cheap and readily to be used. Authors shall discuss further what is the pro and con using the machine learning in the context of logistic and cost. 

Authors shall further discuss more about the advantage, implication and feasibility of machine learning especially in ED (relevant to the title).

The authors shall include studies on meta-analysis for qSOFA, SIRS and MEWS as references.

Author Response

Response to Reviewer 1 Comments These are the comments for the authors: Method section Q1) How authors justify which hospital’s data set to be used as test and validation set? Since it is commonly knowing that for big data set analyses, a ratio of 75:25 training and validation set are good enough. Could the authors explained the rational of using 9:1 ratio for training and validation set? A1) Thank you for your thoughtful comment. Per your comment, traning and validation sets are divided at a ratio of 3:1 or 4:1. If the ratio of the validation set increases, it is difficult to learn enough due to the excessive data loss from the training set. Our data set is divided into three datasets. We divided the dataset into a training-validation set and a test set to verify that the model was overfitting with the training set. The test set consisted of data from other hospitals that were not associated with training and validation to test the model. The training–validation dataset (three hospitals, n = 19,353) and test dataset (one hospital, n = 4,234). The training–validation dataset was divided into the training and validation sets at a 9:1 ratio. Therefore, the ratio of overall dataset is 74% of training set, 8% of validation set and 18% of test set, respectively. We have revised the manuscript as follow: Materials and Methods (revised manuscript page 8, lines 9–17) We divided the dataset into training and validation test sets to prevent the model from overfitting the test set. The test set consisted of data from other hospitals that were not associated with training-validation to test the model. The data from three hospitals (Dongtan Sacred Heart Hospital, Kangnam Sacred Heart Hospital, and Hallym University Sacred Heart Hospital) were used for training-validation (n = 19,353) and the data from one hospital (Chuncheon Sacred Heart Hospital) as the test dataset (n = 4,234). The training-validation set was divided into the training and validation sets at a 9:1 ratio. The overall dataset was divided as follows: 74% for the training set, 8% for the validation set, and 18% for the test set. Q2) Author stated that the AUROC value is higher for machine-learning compare comparing the conventional scoring system by number. The authors shall provide p-value for each comparison of AUROC value against each scoring system to further validate the result. A2) Per your and reviwer #3’s comments, we compared the AUROC by DeLong’s test between qSOFA based machine learning model and conventional qSOFA. Prediction performance of the qSOFA based machine learning model (0.86 [95%CI 0.85–0.87]) for 3-day mortality was significantly higher than the conventional qSOFA scores (0.78 [95%CI 0.77–0.79], p

Reviewer 2 Report

The quick-SOFA is recommended by the most recent sepsis guidelines to identify septic patients at the emergency department. The authors investigated the eligibility of the score to predict the short-term mortality in patients with suspected infection. Indeed, qSOFA was originally build to identify septic patients who usually have a very high hospital mortality (up to 50%). The authors made use of different machine learning models to use the qSOFA for mortality prediction and compared the AUROC with ML-models using SIRS criteria (which have been recommended by former sepsis guidelines) and MEWS.
For training the models, the authors made use of sufficient data from three different emergency departments and the evaluation of the models was done with independent data from a fourth emergency department in South Korea.

The authors presented their results in an intelligible fashion and the paper is well written. Although some details are missing, as outlined below, the paper contains necessary data description and supporting information on the methods to allow reproducibility. The paper is well connected to the ongoing work in the field, fits to the journals’ scope and would be of interest to the readership.

Besides the good overall impression, I have some questions and remarks which needs to be addressed and I see some changes in the methods to further improve the paper:

Abstract:

  • 21: What do you mean with short-term mortality? Is this in-hospital mortality. or 3-day mortality, or both?
  • 25: Please name the ML methods you used in the abstract.
  • 28: please specify: ‘interquartile range’

Introduction:

The introduction is written concisely and motivates the study objective.

  • 41: Please remove ‘new’; thinking of reading the article after the Sepsis-4 guidelines (coming within the next 10 years)
  • 42: I won’t place a comma after simplified

Material and methods:

  • 74: This is one key point of the further analysis and for generalization of the results: How was the suspected infection defined? Was this always documented by the attending physician? Have you made use of reasons for taking blood cultures?
  • 86: ‘white blood cell count’
  • 90: please write ‘body temperature’. Do you distinct how the temperature was measured (invasive/non-invasive / ear, axillary, forehead,…)?
  • 91: It’s good to refer to the valid ranges by Kwon et al., but 30 to 45°C is very loose. I won’t trust a value of 31.3 or 44.6°C – it’s out of human physiology. We use 32° and 43°C as cut-offs. Can you add numbers of how many values you have excluded and how many values are missing at all?
  • 93: How do you assessed the altered mental status? Was it mandatory to fill this information in the EHR (electronic health record)?
  • 97: ‘white blood cell count’, ‘immature band cells’
  • 110: It was a good decision to leaf out the fourth ED to have an independent test set. Have you tried to train on three other EDs and using e.g. Kangnam Sacred Heart Hospital as a test set? I think you should train on all four combinations of test data (leaving out always one ED) and averaging the AUROCs. That give more information on the generalizability of the model. Till now, Chuncheon Sacred Heart Hospital could have a similar patient and care structure than a hospital used for training.
  • 116: Have you oversampled to a 1:1 ratio?
  • 119: 10-fold CV?
  • 121: bracket opening missing
  • 122ff: Please specify the packages by adding the release numbers

How do you handled missing values? E.g. when blood pressure is missing? Do you then assume that hypertension is not present – so you assume the subscore is zero? It is also not clear, what variables entered the models. Was it just the scores itself or do you also passed gender, age, admission reason and respective subscores to the models? Please name the objective functions.

Results:

  • 127: I recommend to state ‘IQR’ instead of ‘range’
  • 132: Table 1 has empty columns
  • 138ff: Some readers might want to know, if the models are overfitted. Can you therefore add the AUROCs for the training set?
  • 147: In the caption I would add that these scores were computed from the independent test set.
  • 161: Figure1: x-label is covered in subplot A and B.
  • 162ff: No need to write ‘Area under the receiver operating characteristic curve’ after (a), (b), (c), (d)
  • 161: Figure1: I would like to see all four curves in one plot instead of four subplots. It would be much easier to compare how the AUROC is changing with time post admission. The same applies to Figure 2.

Author Response

Reviewer #2: Abstract: Q1) 21: What do you mean with short-term mortality? Is this in-hospital mortality. or 3-day mortality, or both? A1) Thank you for your thoughtful comment. The meaning of short-term mortality is 3-day mortality. We have revised the manuscript. Q2) 25: Please name the ML methods you used in the abstract. A2) Machine learning algorithms including extreme gradient boosting (XGB), light gradient boosting machine (LGBM), and random forest were used. Per your suggestion, we described in the abstract. We have revised the manuscript as follow: Abstract (revised manuscript page 3, lines 10–11) Machine-learning algorithms including extreme gradient boosting, light gradient boosting machine, and random forest were used. Q3) 28: please specify: ‘interquartile range’ A3) Per your suggestion, we described interquartile range. We have revised the manuscript as follow: Abstract (revised manuscript page 3, lines 17) interquartile range Introduction: Q4) 41: Please remove ‘new’; thinking of reading the article after the Sepsis-4 guidelines (coming within the next 10 years) A4) Per your suggestion, we removed the word. Q5) 42: I won’t place a comma after simplified A5) Per your suggestion, we removed the comma and introduction section was revised. Material and methods: Q6) 74: This is one key point of the further analysis and for generalization of the results: How was the suspected infection defined? Was this always documented by the attending physician? Have you made use of reasons for taking blood cultures? A6) Thank you for your thoughtful comments. As we demonstrated in limitation section, the study population was difficult to be determined. In this study, we defined diagnoses with suspected infection based on the previous report by Rodriguez et al. (Comparison of qSOFA with current emergency department tools for screening of patients with sepsis for critical illness). We added the list of infectious disease-related diagnoses in Supplementary appendix 2. Among the patients admitted via the ED, the patient whose main diagnosis was infection related diagnosis was extracted. Unfortunately, we cannot confirm that the attending physician has documented the diagnosis due to the restrospective study destion. However, two authors (pulmonologist and anaesthesiologist) reviewed and examined the patient data for eligibility and ambiguous diagnoses were excluded. We have revised the manuscript as follow: Materials and Methods (revised manuscript page 7, lines 2–3) Among the patients admitted via the ED, the patients whose main diagnoses were infection related diagnosis were extracted. Materials and Methods (revised manuscript page 7, lines 4–6) Two of the authors (pulmonologist and anesthesiologist) reviewed and examined the patient data for eligibility, and ambiguous inclusions were excluded by consensus review (Supplementary appendix 2). Discussion (revised manuscript page 13, lines 4–5) Additionally, we cannot confirm that the attending physician had documented the diagnosis. Supplementary appendix 2. Infection-related diagnoses Q7) 86: ‘white blood cell count’ A7) Per your suggestion, we have revised the manuscript as follows: Materials and Methods (revised manuscript page 6, lines 21) white blood cell count Q8) 90: please write ‘body temperature’. Do you distinct how the temperature was measured (invasive/non-invasive / ear, axillary, forehead,…)? A8) Body temperature was assessed using tympanic ear thermometer. Per your suggestion, we have revised the manuscript as follows: Materials and Methods (revised manuscript page 6, lines 24) Body temperature was assessed using tympanic ear thermometer. Materials and Methods (revised manuscript page 7, lines 7) body temperature Q9) 91: It’s good to refer to the valid ranges by Kwon et al., but 30 to 45°C is very loose. I won’t trust a value of 31.3 or 44.6°C – it’s out of human physiology. We use 32° and 43°C as cut-offs. Can you add numbers of how many values you have excluded and how many values are missing at all? A9) Thank you for your thoughtful comment. Per your comments, the excluded range of body temperature (30-45°C) is not a human physiology. Incomplete data of vital signs including body temperature was 115 cases. In this study, the data was extracted from electronic medical recording. But the data can be misrecorded. We excluded the data. We added flow chart of the study. Figure 1. Flow chart Q10) 93: How do you assessed the altered mental status? Was it mandatory to fill this information in the EHR (electronic health record)? A10) At the emergency department, vitals signs and mental status are assessed in all patients by registered nurses for triage. Mental status was measured by AVPU scale (an acronym from "alert, verbal, pain, unresponsive"). Mandatorily, this information fill in the patient records. For calculation of qSOFA score, we considered altered mental status as an AVPU score of anything less than A. We have revised the manuscript as follows: Materials and Methods (revised manuscript page 6, lines 22–24) Mental status was evaluated by emergency specialist nurses using the Alert, Voice, Pain, Unresponsive (AVPU) scale (Supplementary appendix 1). Materials and Methods (revised manuscript page 7, lines 13–14) We considered altered mental status as non-alert per the AVPU scale [31]. Q11) 97: ‘white blood cell count’, ‘immature band cells’ A11) Per your suggestion, we have revised the manuscript as follows: Materials and Methods (revised manuscript page 7, lines 18) white blood cell count Materials and Methods (revised manuscript page 7, lines 19) immature band cells Q12) 110: It was a good decision to leaf out the fourth ED to have an independent test set. Have you tried to train on three other EDs and using e.g. Kangnam Sacred Heart Hospital as a test set? I think you should train on all four combinations of test data (leaving out always one ED) and averaging the AUROCs. That give more information on the generalizability of the model. Till now, Chuncheon Sacred Heart Hospital could have a similar patient and care structure than a hospital used for training. A12) Thank you for your thoughtful comments. We totally agree with your concern. However, your suggestion has several limitations. Four different prediction models are generated according to datasets. We aimed to develop a prediction model with best performance. Therefore, averaging the AUROCs of four models does not seem to be meaning good performance of the model. In addition, since there is a difference in the amount of training data depending on the size of the hospital, there is a possibility of developing a model that has not been sufficiently trained. In this study, the qSOFA based learning model was tested using data from other hospitals that were not associated with training–validation. Chunchen Sacred Heart Hospital is located relatively different area (small town) from other hospitals and we expected that the patient basal characteristics could be different from other hospitals. As we demonstrated in Table 1, test dataset had quietly different patient characteristics (age, severity scores, mortality and so on). Therefore, the results from the test set is reliable. We have revised manuscript as follows: Results (revised manuscript page 9, lines 20–21) There were significant differences in median age, severity of illness scores, ICU admission, and in-hospital mortality between the test and training-validation datasets. Discussion (revised manuscript page 12, lines 23–24) These results are reliable because the model was tested using an independent test set that had patient characteristics different from those in training-validation. Q13) 116: Have you oversampled to a 1:1 ratio? A13) Yes, it is. We have revised manuscript as follows: Materials and Methods (revised manuscript page 8, lines 23) to a 1:1 ratio Q14) 119: 10-fold CV? A14) No, 5-fold cross-validation. We have revised the manuscript as follows: Materials and Methods (revised manuscript page 9, lines 1–2) We performed 5-fold cross validation on the training dataset and tuned the hyperparameters using grid search. Q15) 121: bracket opening missing A15) A16) Per your suggestion, we have revised the manuscript as follows: Materials and Methods (revised manuscript page 9, lines 3–5) We evaluated models with the ROC curve and AUROC using Anaconda (Python version 3.7, https://www.anaconda.com; Anaconda Inc., TX, USA), Q16) 122ff: Please specify the packages by adding the release numbers A16) Per your suggestion, we described the release numbers. Materials and Methods (revised manuscript page 9, lines 5–7) the XGBoost package version 0.90 (https: //xgboost.readthedocs.io), the LGBM package version 2.2.3 (https://lightgbm.readthedocs.io/en/latest/Python-Intro.html), and the imbalanced-learn package version 0.5.0 (https://imbalanced-learn.readthedocs.io). Q17) How do you handled missing values? E.g. when blood pressure is missing? Do you then assume that hypertension is not present – so you assume the subscore is zero? It is also not clear, what variables entered the models. Was it just the scores itself or do you also passed gender, age, admission reason and respective subscores to the models? Please name the objective functions. A17) As shown in Figure 1 fow chart, incomplete data were excluded. We aimed to comare the conventional qSOFA and qSOFA based machine learnig mode. So, our model was generated using three qSOFA varibles. Results: Q18) 127: I recommend to state ‘IQR’ instead of ‘range’ A18) Per your suggestion, we have revised the manuscript as follows: Results (revised manuscript page 9, lines 15) IQR Q19) 132: Table 1 has empty columns A19) Sorry for the confusion. We revised the Table 1. Table 1. Characteristics of patients with suspected infection in the emergency department Variable Total (n = 23587) Training-validation datasets (n = 19353) Test datasets (n=4234) P value Median age (IQR) 63 (43–78) 62 (42–77) 67 (50–79)

Reviewer 3 Report

The presented paper describes a study where machine learning models were developed and assessed for mortality prediction in patients with suspected infection. My overall impression is that the manuscript does not adequately describe the background, method and material, results and discussion. In addition, the conclusions from the study are not justified by the data as the conclusions are not based on statistical comparisons of the results. 

The Introduction does not provide sufficient background as much more work have been done in this area and more references should be included. What is the novelty of this study and why is it needed? When it comes to the Material and Method section, it is not adequately described as much information is missing. E.g., definition of suspected infections? Insufficient information regarding the work with developing the different machine learning models. Not possible to reproduce the study in the same way by reading the Material and Method.

When it comes to the research design, a better study design would be to mix data from all four hospitals and then randomly assign each patient to training data cohort or test data cohort. Moreover, training-validation data sets are usually set as 3:1 ratio, not 9:1.

Not all results are presented. I would like to see the results obtained from each of the different machine learning algorithms for each outcome variable. 

Moreover, comparisons between AUROCs should be done statistically, otherwise, you cannot draw any conclusions about the results. For example, DeLong´s test can be used if you are comparing two curves that have been generated using the same data but different models/methods.

The figure legends do not contain enough information, it is not possible to understand what the figures are showing by just reading the legends. 

Author Response

Reviewer #3: Q1) The presented paper describes a study where machine learning models were developed and assessed for mortality prediction in patients with suspected infection. My overall impression is that the manuscript does not adequately describe the background, method and material, results and discussion. In addition, the conclusions from the study are not justified by the data as the conclusions are not based on statistical comparisons of the results. A1) Thank you for your careful reviewing our manuscript and giving valuable comments. We have made many corrections and modification following your helpful comments. We believe we have addressed all questions and comments, but please give us more opportunity to respond if there is anything unsatisfactory or questionable. Q2-1) The Introduction does not provide sufficient background as much more work have been done in this area and more references should be included. A2-1) Per your suggestion, we add references of meta-analysis for qSOFA, SIRS and MEWS. We have revised the introduction part as follows: Introduction (revised manuscript page 5, lines 2–13) Early recognition and prompt treatment of sepsis in emergency department (ED) are important to improve the patient outcomes [1]. Sepsis was characterized as a systemic inflammatory response syndrome (SIRS) to infection [2]. The quick Sepsis-related Organ Failure Assessment (qSOFA) criteria [3], was introduced by the Third International Consensus Definitions for Sepsis and Septic Shock (Sepsis-3). The qSOFA scores can be used outside of the intensive care unit (ICU) for predicting mortality or ICU stay [4-9]. However, several meta-analyses of qSOFA scores showed that the scores had a poor sensitivity for predicting in-hospital mortality [10-14]. Early warning scores [15,16], which is a tool for identifying hospitalized patients at risk of deterioration has been proposed to be used for predicting hospital mortality with suspected sepsis in the ED [7,17-20]. Hamilton et al. reported that the early warning scores were not accurate in predicting sepsis mortality in ED (67% of sensitivity and 60% of specificity) [21]. Q2-2) What is the novelty of this study and why is it needed? A2-2) Recently, various machine learning methods for predicting outcomes more accurately have been implemented in the medical field [22-24]. Machine learning models for the early identification of patients at risk for sepsis have been developed in the ICU [25] and ED settings [26-28]. Although these diverse machine-learning models can improve predictive accuracy for sepsis outcomes, they require too many variables and laboratory results that may not be available in the ED. These factors can lead to poor generalizability of the machine-learning-based prognostic models. We sought to develop a machine learing based prognostic model with simple variables and generally applicable. We have revised the introduction part as follows: Introduction (revised manuscript page 5, lines 14–20) Recently, various machine-learning methods for predicting outcomes more accurately have been implemented in the medical field [22-24]. Machine-learning models for the early identification of patients at risk for sepsis have been developed in the ICU [25] and ED settings [26-28]. Although these diverse machine-learning models can improve predictive accuracy for sepsis outcomes, they require excessive variables and laboratory results that may not be available in the ED. These factors can lead to poor generalizability of the machine-learning-based prognostic models. Q2-3) When it comes to the Material and Method section, it is not adequately described as much information is missing. E.g., definition of suspected infections? A2-3) Thank you for your thoughtful comments. As we demonstrated in limitation section, the study population was difficult to be determined. In this study, we defined diagnoses with suspected infection based on the previous report by Rodriguez et al. (Comparison of qSOFA with current emergency department tools for screening of patients with sepsis for critical illness). We added the list of infectious disease-related diagnoses in Supplementary appendix 2. Among the patients admitted via the ED, the patient whose main diagnosis was infection related diagnosis was extracted. Unfortunately, we cannot confirm that the attending physician has documented the diagnosis due to the restrospective study destion. However, two authors (pulmonologist and anaesthesiologist) reviewed and examined the patient data for eligibility and ambiguous diagnoses were excluded. We have revised the manuscript as follow: Materials and Methods (revised manuscript page 7, lines 2–3) Among the patients admitted to the ED, the data of those whose main diagnoses were infection-related were extracted. Materials and Methods (revised manuscript page 7, lines 4–6) Two of the authors (pulmonologist and anesthesiologist) reviewed and examined the patient data for eligibility, and ambiguous inclusions were excluded by consensus review (Supplementary appendix 2). Discussion (revised manuscript page 13, lines 2–3) Additionally, we cannot confirm that the attending physician had documented the diagnosis. Supplementary appendix 2. Infection-related diagnoses Q2-4) Insufficient information regarding the work with developing the different machine learning models. Not possible to reproduce the study in the same way by reading the Material and Method. A2-4) Sorry for the confusion. Per your comment, we added more information of the work for developing machine learning models. We have revised the manuscript as follow: Materials and Methods (revised manuscript page 8, lines 7–page 9, lines 9) The qSOFA-based machine-learning model was developed using the qSOFA criteria, including systolic blood pressure, respiratory rate, and mental status. A dataset was created with the variables of the 23,587 patients. We divided the dataset into training and validation test sets to prevent the model from overfitting the test set. The test set consisted of data from other hospitals that were not associated with training-validation to test the model. The data from three hospitals (Dongtan Sacred Heart Hospital, Kangnam Sacred Heart Hospital, and Hallym University Sacred Heart Hospital) were used for training-validation (n = 19,353) and the data from one hospital (Chuncheon Sacred Heart Hospital) as the test dataset (n = 4,234). The training-validation set was divided into the training and validation sets at a 9:1 ratio. The overall dataset was divided as follows: 74% for the training set, 8% for the validation set, and 18% for the test set. Datasets were standardized by min-max scaling. Regarding the model algorithm, extreme gradient boosting (XGB), light gradient boosting machine (LGBM), and random forest were used. In the training dataset, deaths and ICU inpatients were much fewer than survivors and general ward inpatients, respectively. This data imbalance can bias the machine-learning models and render them inaccurate. To solve this problem, we trained the model after balancing the training dataset through the synthetic minority oversampling technique to a 1:1 ratio in the XGB and LGBM models and modeled the balanced random forest. Three models were trained with the basic hyperparameters and training sets and evaluated with a validation set to select the one with the best performance. We performed 5-fold cross validation on the training dataset and tuned the hyperparameters using grid search. After that, tuning was performed to obtain an optimal hyperparameter through 5-fold cross-validation. The final model was validated through the test set. We evaluated models with the ROC curve and AUROC using Anaconda (Python version 3.7, https://www.anaconda.com; Anaconda Inc., TX, USA), the XGBoost package version 0.90 (https: //xgboost.readthedocs.io), the LGBM package version 2.2.3 (https://lightgbm.readthedocs.io/en/latest/Python-Intro.html), and the imbalanced-learn package version 0.5.0 (https://imbalanced-learn.readthedocs.io). DeLong’s test was used to compare AUROCs between qSOFA scores and qSOFA-based machine-learning models [33]. Q3) When it comes to the research design, a better study design would be to mix data from all four hospitals and then randomly assign each patient to training data cohort or test data cohort. Moreover, training-validation data sets are usually set as 3:1 ratio, not 9:1. A3) Thank you for your thoughtful comment. Per your comment, traning and validation sets are divided at a ratio of 3:1 or 4:1. If the ratio of the validation set increases, it is difficult to learn enough due to the excessive data loss from the training set. Our data set is divided into three datasets. We divided the dataset into a training-validation set and a test set to verify that the model was overfitting with the training set. The test set consisted of data from other hospitals that were not associated with training and validation to test the model. Chunchen Sacred Heart Hospital (test set) is located relatively different area (small town) from other hospitals and we expected that the patient basal characteristics could be different from other hospitals. As we demonstrated in Table 1, test dataset had quietly different patient characteristics (age, severity scores, mortality and so on). Therefore, the results from the test set is reliable. The training–validation dataset (three hospitals, n = 19,353) and test dataset (one hospital, n = 4,234). The training–validation dataset was divided into the training and validation sets at a 9:1 ratio. Therefore, the ratio of overall dataset is 74% of training set, 8% of validation set and 18% of test set, respectively. We have revised the manuscript as follow: Materials and Methods (revised manuscript page 8, lines 9–17) We divided the dataset into training and validation test sets to prevent the model from overfitting the test set. The test set consisted of data from other hospitals that were not associated with training-validation to test the model. The data from three hospitals (Dongtan Sacred Heart Hospital, Kangnam Sacred Heart Hospital, and Hallym University Sacred Heart Hospital) were used for training-validation (n = 19,353) and the data from one hospital (Chuncheon Sacred Heart Hospital) as the test dataset (n = 4,234). The training-validation set was divided into the training and validation sets at a 9:1 ratio. The overall dataset was divided as follows: 74% for the training set, 8% for the validation set, and 18% for the test set. Discussion (revised manuscript page 12, lines 23–24) These results are reliable because the model was tested using an independent test set that had patient characteristics different from those in training-validation. Q4) Not all results are presented. I would like to see the results obtained from each of the different machine learning algorithms for each outcome variable. A4) Per your suggestion, we added the results from each of the different machine learning algroithms in Supplementary appendix 3 as follows: Supplementary appendix 3. Predictive performance of algorithm-specific machine learning models through validation sets Algorithm 3-day mortality In-hospital mortality 3-day ICU admission ICU admission qSOFA score ≥ 2 qSOFA score < 2 BRF 0.85 (0.79–0.86) 0.78 (0.75–0.79) 0.77 (0.75–0.78) 0.75 (0.73–0.75) 0.67 (0.57–0.75) 0.84 (0.79–0.87) XGB 0.76 (0.74–0.79) 0.76 (0.76–0.78) 0.79 (0.79–0.79) 0.78 (0.78–0.79) 0.61 (0.58–0.65) 0.79 (0.78–0.83) LGBM 0.70 (0.69–0.73) 0.71 (0.69–0.73) 0.78 (0.77–0.78) 0.75 (0.74–0.75) 0.60 (0.56–0.65) 0.68 (0.64–0.71) BRF=balanced random forest; XGB=extreme gradient boosting; LGBM=light gradient boosting machine; ICU=intensive care unit; qSOFA=quick Sepsis-related Organ Failure Assessment score. Q5) Moreover, comparisons between AUROCs should be done statistically, otherwise, you cannot draw any conclusions about the results. For example, DeLong´s test can be used if you are comparing two curves that have been generated using the same data but different models/methods. A5) Per your comments, we compared the AUROC by DeLong’s test between qSOFA based machine learning model and conventional qSOFA. Prediction performance of the qSOFA based machine learning model (0.86 [95%CI 0.85–0.87]) for 3-day mortality was significantly higher than the conventional qSOFA scores (0.78 [95%CI 0.77–0.79], p
